# Application of Xanthan-Gum-Based Edible Coating Incorporated with *Litsea cubeba* Essential Oil Nanoliposomes in Salmon Preservation

**DOI:** 10.3390/foods11111535

**Published:** 2022-05-24

**Authors:** Haiying Cui, Mei Yang, Ce Shi, Changzhu Li, Lin Lin

**Affiliations:** 1School of Food and Biological Engineering, Jiangsu University, Zhenjiang 212013, China; cuihaiying@ujs.edu.cn (H.C.); kjld2008@126.com (M.Y.); ceshi@ujs.edu.cn (C.S.); 2State Key Laboratory of Utilization of Woody Oil Resource, Hunan Academy of Forestry, Changsha 410007, China

**Keywords:** edible coating, *Litsea cubeba* essential oil, nanoliposome

## Abstract

Salmon is prone to be contaminated by *Vibrio parahaemolyticus* (*V. parahaemolyticus*), leading to the deterioration of salmon quality and the occurrence of food-borne diseases. In this study, we aimed to develop a novel xanthan-gum-based edible coating embedded with nano-encapsulated *Litsea cubeba* essential oil (LC-EO) for salmon preservation at 4 °C. First, the results of the growth curves and scanning electron microscopy (SEM) showed that LC-EO displayed potent antibacterial activity against *V. parahaemolyticus*; the optimal concentration of LC-EO in the liposomes was 5 mg/mL, and the maximal encapsulation efficiency (EE) was 37.8%. The particle size, polydispersity coefficient (PDI), and zeta potential of the liposomes were 168.10 nm, 0.250, and −32.14 mV, respectively. The rheological test results of xanthan-gum-based edible coatings incorporating liposomes showed that the prepared coating was suitable for applying on food surfaces. The results in the challenge test at 4 °C demonstrated that the treatment of 1:3 (liposome: xanthan gum, *v*/*v*) coating performed the best preservative properties, the coating treatment delayed the oxidation of salmon, and controlled the growth of *V. parahaemolyticus*. These findings suggest that the coatings formulated in this study could be used as a promising approach to control *V. parahaemolyticus* contamination and maintain salmon quality.

## 1. Introduction

Salmon, which enjoys the reputation of “treasures in water”, has become a popular raw aquatic product for consumers because of its taste, rich nutrients [1], and its potential ideal preventive effect on chronic diseases [2,3]. However, the plentiful free amino acids and unsaturated fatty acids in salmon are prone to protein and lipid oxidation under the action of endogenous enzymes, oxygen, and other factors during the refrigeration process, causing discoloration [4]. Salmon is also easily contaminated by *V. parahaemolyticus* during farming, transportation, storage, sales, and edible processing [5], leading to the occurrence of food-borne diseases, such as diarrhea, headache, nausea, vomiting, and abdominal cramps [6], which has become an increasingly serious public health problem in the whole world, including China [7].

In view of the negative effects of chemical food additives and antibiotics [8], new preservative technologies developed with natural bioactive substances possess great application potential in the food industry [9]. In recent years, essential oils (EOs) derived from plants have attracted extensive attention due to the safe, antioxidant and antibacterial properties [10].

As the secondary product extracted from the fresh fruit of *Litsea cubeba*, *Litsea cubeba* essential oil (LC-EO) has been disclosed to have a good inhibitory effect on various food-borne pathogens, but few research studies have reported on *V. parahaemolyticus* yet [11]. In addition, LC-EO is believed to be impractical to directly apply to food products due to some drawbacks, such as the low effectiveness, unpleasant smell, and some negative effects on food. Nanoliposome technology is a promising method to overcome these challenges. This can be credited to its ability to encapsulate EOs within the cavity formed by the hydrophilic head and hydrophobic tail of liposomes [12]. In this way, not only can the drawback of the chemical instability of EOs be overcome, but also possible adverse interactions with food ingredients can be limited [13,14]. However, the direct application of liposomes on foods such as salmon may cause aggregation or rupture, leading to the leakage of encapsulated substances [15]. Therefore, edible coatings as the carrier of nanoliposomes might be considered. Among various source materials of edible coating, polysaccharides, proteins, and lipids are favored for their cost effectiveness. Additionally, the environmental friendliness compared to chemically synthesized packaging materials such as polyethylene is also a vital element [16]. In addition, the application of edible coatings incorporating antimicrobial and antioxidant agents on food packaging can not only perform the basic functions of traditional packaging, but also further delay food spoilage and oxidation [17,18]. Chen et al. used a chitosan coating combined with oregano or cinnamon essential oil in roast duck to control the proliferation of microorganisms and the oxidation of food ingredients [19]. In general, polysaccharides, such as chitosan, pectin, and carrageenan, are widely used as the materials of edible coating. Xanthan gum (XG), an exopolysaccharide produced by the fermentation of Xanthomonas, is a promising material of edible coating due to the exceptional stability and high viscosity [20]. 

Currently, few studies on the incorporation of nano-encapsulated bioactives in edible coatings for food preservation have been reported. Therefore, this study aimed to prepare and characterize the LC-EO-loaded nanoliposomes, and then incorporate them into edible coatings based on XG. Finally, the effects of coating treatment on the physicochemical properties of salmon, and the growth of *V. parahaemolyticus* at 4 °C were investigated to provide a new reference for the field of raw aquatic product preservation.

## 2. Materials and Methods

### 2.1. Materials and Bacterial Culture 

LC-EO was purchased from J. E International (Caussols plateau, France); soy lecithin and cholesterol were obtained from China National Medicine Co., Ltd. (Beijing, China); polyvinylpyrrolidone (PVP) was provided by Sinopharm Chemical Reagent Co., Ltd. (Shanghai, China); XG was purchased from Shanghai Macklin Biochemical Co., Ltd. (Shanghai, China); *V. parahaemolyticus* ATCC17802 was bought from China Center of Industrial Culture Collection (Beijing, China) and cultivated in a tryptic soy broth (TSB) medium containing 3% NaCl at 37 °C for 48 h with shaking.

### 2.2. Antibacterial Activity of LC-EO

The susceptibility test was carried out based on the methods described by previous research [21,22]. LC-EO was suspended into the tubes of TSB medium containing 3% NaCl to obtain the various concentrations of 0.0625, 0.125, 0.25, 0.5, and 1 mg/mL, respectively, by the double dilution method. Then the bacterial suspension at the concentration of 10^5^ CFU/mL with or without LC-EO (served as control) was added into the new tubes and cultured at 37 °C for 48 h with shaking. The minimum inhibitory concentration (MIC) was defined as the lowest concentration of LC-EO at which *V. parahaemolyticus* growth could not be observed. Subsequently, the bacterial suspension that contained the concentration of LC-EO was not lower than MIC was streaked on the solid medium. The lowest LC-EO concentration at which no colony growth was observed on the medium was the minimum bactericidal concentration (MBC).

The growth curves of *V. parahaemolyticus* exposed to LC-EO was determined according to the method described by Fan et al. [23] with appropriate modifications. *V. parahaemolyticus* was cultured in a TSB medium containing 3% NaCl for 12 h, and the optical density at 600 nm (OD_600 nm)_ of the bacterial suspension was adjusted to about 0.50 (10^8^ CFU/mL). Then, 1% of the inoculum was added into the 3% NaCl TSB medium containing 0, 1/4 MIC, 1/2 MIC and MIC concentrations of LC-EO. The final concentration of bacterial suspension used in this test was 10^5^ CFU/mL. At the same time, the TSB medium containing 3% NaCl inoculated with 0, 1/4 MIC, 1/2 MIC, and MIC concentration of LC-EO but without bacteria was set as the control group to eliminate the impact of LC-EO on the OD_600 nm_ values. The samples were incubated at 37 °C with shaking, and the OD_600 nm_ was measured within 24 h to plot the growth curves.

The effect of LC-EO treatment on the morphology and structure of *V. parahaemolyticus* was observed by SEM (COXEM EM-30 Plus, Daejeon, Korea).

### 2.3. Preparation and Characterization of LC-EO Nanoliposomes

#### 2.3.1. Preparation of LC-EO Nanoliposomes

LC-EO nanoliposomes were prepared by the thin film dispersion method according to a previous study with slight modifications [24]. Soy lecithin (0.4 g), cholesterol (0.08 g, 5:1, *w*/*w*), and LC-EO (5 mg/mL) were added into 20 mL trichloromethane. The mixture was ultrasonically dissolved in an ultrasonic cleaner, and then placed in a rotary evaporator to evaporate. Then the organic solvent was removed until a smooth lipid film was formed. The film was dried in a vacuum drying oven at 30 °C for 24 h, following by the addition of PVP (0.02 g) and phosphate buffered saline (20 mL). The uniform suspension was fragmented in a cell ultrafine grinder with 360 W for 15 min to obtain LC-EO nanoliposomes. 

#### 2.3.2. Characterization of LC-EO Nanoliposomes

##### Determination of Particle Size, PDI, Zeta Potential, and EE of LC-EO

The Particle Size Analyzer (Nano ZS90, Malvern Instruments, Worcester, UK) was used for measuring the particle size, PDI and zeta potential of nanoliposomes [25].

After centrifuging the liposomes, the absorbance of the supernatant was measured at 225 nm (maximum absorption wavelength of the main component of LC-EO), and the EE of LC-EO was calculated by the following formula [26]: EE% = (Total LCEO − Free LCEO)/Total LCEO × 100%

##### Fourier Transform Infrared Spectroscopy (FTIR)

FTIR (Thermo Nicolet Corporation, Waltham, MA, USA) was applied to study the molecular interactions between LC-EO and liposomes in the wavelength range of 400 to 4000 cm^−1^. The background scan was performed before the sample measurement, and all samples were scanned 32 times [27].

##### SEM

The prepared LC-EO liposomes were appropriately diluted and dropped on the silicon wafer. After natural air drying, the morphology of liposomes was observed by SEM.

### 2.4. Preparation of Coating Solutions

The XG solution (5 mg/mL) was prepared by dissolving XG in distilled water under a magnetic stirrer for 2 h at 55 °C. The XG solution and the previously prepared liposome solution were mixed at 1:0, 1:1, 1:2, 1:3, 1:4, and 1:5 (*v*/*v*), respectively. Then the mixture was gently stirred for 30 min to obtain six different coating solutions.

### 2.5. Rheological Properties of the Coating Solutions

A rheometer (DHR-1, Waters, MA, USA) was used for rheological analysis of coating solutions at 25 °C with shear rates varying from 0 to 200 s^−1^. The distance between parallel stainless-steel plates (40 mm in diameter) was set to 2 mm, and the time interval between sample loading and actual measurement was 30 min [28]. The power–law model was employed to study the variation of coating solution viscosity, and the shear stress with shear rate was calculated as follows:σ = k × γ^n^
where σ is the shear stress (Pa), k is the consistency coefficient (Pa⋅s), γ is the shear rate (s^−1^), and n is the flow behavior index.

### 2.6. Challenge Test of Edible Coating in Salmon Preservation

#### 2.6.1. Coating Process of Salmon Fillet

Fresh salmon fillets were cut into slices in the same size (3 × 3 × 2 cm) and randomly divided into 7 groups, namely control (coated with distilled water), XG (xanthan gum, coated with XG solution) and XG and liposome with different ratio (1:1, 1:2, 1:3, 1:4 and 1:5, respectively). Each group was carried out in triplicate. The salmon slices were immersed in different coating solutions for 60 s, air-dried at room temperature, then placed in a glass plate and stored at 4 °C for up to 8 days. 

#### 2.6.2. pH

The salmon sample (5 g) was mixed with distilled water at a 1:9 ratio and then homogenized by a homogenizer at 8000 rpm for 2 min. The pH of the homogenate was immediately measured after filtration, and each measurement was repeated three times for each sample. 

#### 2.6.3. Lipid Oxidation

The amount of thiobarbituric acid active substances (TBARS) produced was used to evaluate the degree of lipid oxidation of salmon samples [29]. A 10 g sample was blended in 50 mL trichloroacetic acid solution (contains 7.5% trichloroacetic acid and 0.1% ethylenediaminetetraacetic acid) and homogenized at 8000 rpm for 2 min. Then, 5 mL of filtrate from the above mixture was reacted with 5 mL thiobarbituric acid solution (0.02 mol/L), and then the whole reaction solution was kept at 100 °C water bath for 30 min [30]. After cooling down, the absorbance of the solution was determined at 532 nm by UV–Vis spectrophotometer (UV-1601). The TBARS values were calculated by comparing with the standard curve of 1,1,3,3-tetraethoxypropane and expressed as milligram malondialdehyde (MDA) equivalent per kilogram salmon product (mg MDA/kg).

#### 2.6.4. Protein Oxidation

Ellman reagent 5,5′-dithio-bis (2-nitrobenzoic acid) (DTNB) was applied to determine the content of free sulfhydryl groups in salmon samples and evaluate the degree of protein oxidation [31]. A 2.0 g salmon sample and 50 mL of Tris (0.1 M, pH = 8) buffer containing 5% sodium dodecyl sulfate (SDS) was blended, homogenized, and heated in a constant temperature water bath at 80 °C for 30 min. Then, 5% SDS Tris (0.1 M pH = 8) buffer was used to adjust the protein in the filtrate to an appropriate concentration. Subsequently, 0.5 mL diluted filtrate, 2.0 mL Tris buffer (0.1 M pH = 8) and 0.5 mL 10 mM DTNB in 0.1 M Tris buffer (pH = 8) were mixed and incubated in the dark for 30 min. Then the absorbance was measured at 412 nm wavelength, and the free sulfhydryl content was calculated with a molar extinction coefficient of 13,600 L·mol^−1^·cm^−1^ [32]. The eventual experimental results were presented in the form of nmol sulfhydryl groups/mg protein, where the protein content was determined by using bovine serum albumin as standard substances.

#### 2.6.5. Surface Color

As a quality parameter of salmon, the surface color was evaluated by a colorimeter (Color Quest XE, Reston, VA, USA). The salmon samples with different coating treatment were stored at 4 °C at 0, 2, 4, 6, 8 days, respectively. Then the values of lightness (L*), redness (a*), and yellowness (b*) were determined at different time points. Each sample was measured in parallel three times for statistical analysis.

#### 2.6.6. Texture Evaluation

Samples coated with distilled water were regarded as the control groups, and the measurement of the salmon texture was progressed with a texture analyzer with the P/50 probe (TA-XT2i, Stable Micro Systems Ltd., Godalming, UK). Salmon samples were compressed twice with 30% compression, and the experimental results were exported for data analysis using EXPONENT software [33].

#### 2.6.7. Microbial Analyses

To investigate the antimicrobial activity of the prepared coating, the total viable count (TVC) microorganisms in salmon samples with or without coating treatment was determined using plate count agar.

To assess the inhibitory effect of the prepared coating on *V. parahaemolyticus*, the treatment effect of 1:3 coating on the growth of *V. parahaemolyticus* in salmon was assessed by the following method. Fresh salmon samples were cut into slices of the same size (around 4 g) and sterilized with UV light for 30 min. Afterward, samples were immersed in *V. parahaemolyticus* suspension (~10^3^ CFU/mL) for 30 min and naturally dried in a biological safety cabinet. Then, the samples were uniformly sprayed with prepared coating solution and stored at 4 °C for 8 days. The salmon slices without coating served as the control group, and results were expressed as the amount of microorganisms (CFU/g). Three parallel groups were set in each group for statistical analysis. 

### 2.7. Statistical Analysis

Each experiment was carried out in triplicate, and the data were recorded as mean ± standard deviation (SD). Comparisons between sample means were performed by one-way ANOVA using SPSS software 22.0 (IBM Corp., Armonk, NY, USA). Statistical differences between data were identified when *p* < 0.05.

## 3. Results and Discussion

### 3.1. Antibacterial Activity of LC-EO against V. parahaemolyticus

The MIC and MBC values of LC-EO against *V. parahaemolyticus* were 0.25 mg/mL and 0.5 mg/mL, respectively. Results in Figure 1a show the growth curves of *V. parahaemolyticus* with or without LC-EO treatment. The growth of *V. parahaemolyticus* without LC-EO treatment tended to be stable after incubation for 6 h, while the lag phase of bacteria treated with LC-EO was prolonged, the maximum specific growth rate decreased, and the impact was gradually enhanced with the increasing concentrations of LC-EO. 

The morphological changes of *V. parahaemolyticus* induced by LC-EO were observed by SEM. Results revealed that the bacterial cells in the control group were short rod-shaped or slightly curved arc-shaped with a plump shape and smooth surface (Figure 1b). In contrast, shrinkage and depression occurred in the cells after treatment with LC-EO (Figure 1c). These results confirmed the irreversible destruction of *V. parahaemolyticus* cells induced by LC-EO. 

### 3.2. Characterization of LC-EO Liposomes

The characterizations of the prepared liposomes loading varied concentrations of LC-EO are exhibited in Table 1. The particle size of nanoliposome in the control group was 101.27 nm, while with the increasing concentration of LC-EO treatment, the mean diameters of the liposomes increased from 149.92 nm to 185.39 nm, and the PDI values used for characterizing the dispersibility between particles were all within the range of 0.20 and 0.30, indicating that the liposome particles were evenly distributed [34]. Generally, dispersed systems with high zeta potential (>|30| mV) are expected to be relatively stable, which can be ascribed to the mutual repulsion of charged particles [35]. In this case, the unavoidable aggregation tendency between particles was overcome, and the system stabilization was achieved [36]. The encapsulation efficiency of LC-EO is one of the important indicators for the evaluation of nanoparticle quality because of its relation to the practical application effect of nanoparticles. The maximal LC-EO encapsulation efficiency of nanoliposomes was attained as 37.8%, with 5 mg/mL LC-EO being coated. Based on the above analysis, nanoliposomes loaded with 5 mg/mL LC-EO were selected for subsequent experimental materials due to their suitable diameters, high zeta potential, and encapsulation efficiency.

From the results in Figure 2b,c, the prepared liposomes performed uniform spherical structures. In addition, the diameter of LC-EO-loaded liposomes was larger than that of LC-EO-unencapsulated liposomes, which was consistent with the above results.

The FTIR spectra of the blank liposomes, LC-EO, LC-EO liposomes are shown in Figure 2a. The absorption peak of LC-EO at 1672 cm^−1^ was attributed to carbonyl (C=O) stretching vibration, and the peaks at 1444 cm^−1^ and 1378 cm^−1^ represented the C=C vibration and CH_3_ mixed vibration, respectively [37]. The 2921 cm^−1^ and 2855 cm^−1^ featured stretching vibration of CH_3_ and CH_2_ [38]. The spectral pattern of LC-EO liposomes almost corresponded with that of blank liposomes, although the characteristic absorption intensities at 2926, 2856, 1466, and 1374 cm^−1^ were enhanced. However, the characteristic absorption intensities of blank liposomes were weaker than that of LC-EO. Additionally, the disappearance of the characteristic peak at 1672 cm^−1^ also affirmed the successful embedment of LC-EO into liposomes.

### 3.3. Rheological Analysis of Coating Solutions

Figure 3 depicts the variation law of the viscosity and shear stress of coating solutions with the shear rate. In the experiment, the viscosity of each coating solution decreased, and the shear stress increased as the shear rate increased, which was a characteristic shear-thinning behavior for pseudoplastic fluid [39]. Under the action of intramolecular and intermolecular non-covalent bonds, the entanglement between the molecular segments in the XG solution forms a rigid structure, resulting in the high viscosity of the XG solution at low shear rates, which contributes to its excellent particle suspension properties [40]. However, considering the uniformity of the coating solution distribution on the food surface and the ideal coating thickness, a viscosity lower than 0.7 Pa is considered desirable [41]. The experimental results (shown in Table 2) revealed that coating solutions prepared in this study equipped suitable viscosity, which would facilitate the formation of the uniform film on the food surface.

### 3.4. Salmon Preservation by Edible Coating

#### 3.4.1. pH

The changes of the pH value of fish products during storage can be used as a vital indicator for their quality evaluation. Salmon samples in this study presented an initial pH of 6.22, indicating the freshness of samples (Figure 4). The changing trend of pH among each group was similar; the pH values decreased during the initial 4 days first and then increased until the 8th day. The initial decrease was speculated to be resulted from the accumulation of lactic acid during glycolysis, which was similar to meat products after post-slaughter [42]. The pH increase could be attributed to the accumulation of basic nitrogen compounds, such as ammonia and trimethylamine, caused by the hydrolysis of proteins, fats, and other substances in fish samples by endogenous and exogenous enzymes [43]. It could be concluded from the result that coating treatments, except the control group, slowed down the pH changes of samples, to be specific, the decrease in the first four days and the increase in the last four days. When compared with coating groups incorporating LC-EO liposomes, the pH of XG coating group experienced greater initial decrease and followed an increase on each measurement day. It might be related to the lack of substances with antioxidant and antibacterial activities. The positive effect of coating solution on salmon quality was not strictly consistent with nanoliposome concentration changes in the coating.

#### 3.4.2. TBARS Determination

Salmon is rich in unsaturated fatty acids, which makes it prone to lipid oxidation and, subsequently, quality deterioration. The TBARS value was a commonly used indicator to evaluate lipid oxidation degree [44]. It was figured by measuring the absorbance value after the reaction of mixing lipid secondary oxidation products with 2-thiobarbituric acid. The initial TBARS value was 0.27 mg MDA/kg (Figure 5), which was much lower than the international upper limit threshold for fish: 1–2 mg MDA/kg [45], indicating the freshness of materials. 

From the results in Figure 5, a significant difference in the degree of lipid oxidation between the control group and the coating groups was observed (*p* < 0.05). In control group, the TBARS value reached its peak, exceeding the threshold (1 mg MDA/kg) on the 6th day, revealing severe deterioration among these samples. Compared with the control group, the other coating treatments all showed an inhibitory effect on lipid oxidation, and the results showed that the lipid oxidation progress of the samples in the 1:3 and 1:4 coating treatment groups was the slowest. In addition, the TBARs values of the samples in the XG group were significantly larger than those in the other liposome-containing coating groups (*p* < 0.05). This indicated that the single XG coating could form a barrier on the contact interface between fat and oxygen to inhibit fat oxidation, but its positive effect on fat oxidation was limited. Similar to the experimental results reported by Kulawik et al. [46], the TBARS values among different groups experienced a decline at the end of the experiment, which could be the oxidation products of fatty acids interacting with other fish components, such as amino acids or breaking down into other substances, or a combination as mentioned in the above two situations [47]. 

#### 3.4.3. Protein Oxidation

The decrease in free sulfhydryl groups was one of the critical implications of protein oxidation in fish. Sulfhydryl groups of amino acids, such as cysteine, exposed on the surface of proteins are susceptible to oxidation, resulting in the formulations of intermolecular or intramolecular disulfide bonds, which makes it possible to imply the degree of protein oxidation through changes in sulfhydryl content [48].

The changes of free sulfhydryl content in salmon samples under different treatments at 4 °C are illustrated in Figure 6. After 8 days of storage, the content of free sulfhydryl groups in all groups significantly decreased (*p* < 0.05). In addition, the free sulfhydryl in control group and XG group experienced similar substantial reductions. It was suspected that sulfhydryl groups in samples from control and XG groups were constantly exposed under air, and fully combined with oxygen to form disulfide bonds, leading to the violent oxidation of proteins. With the prolongation of the storage period, it was worth noting that the difference in free sulfhydryl content between the coating groups containing liposomes and the non-added liposomes coating groups was gradually significant (*p* < 0.05), and the 1:3 coating treatment group was the most noticeable. This was comparable to the result presented by Zhao et al. [49], who reported that the protective effect of active additives on sulfhydryl groups did not always increase with increasing concentration of active substance. The protective effect on sulfhydryl groups might be the result of the antioxidant capacity of the abundant aldehydes contained in LC-EO. Existing studies have also mentioned the similar protective effect of protein oxidation when natural supplements, such as resveratrol and grape seed extract, are applied in food [50].

#### 3.4.4. Surface Color

After salmon acquires pigments, such as astaxanthin, by preying on aquatic animals, the pigments bind to the surface of proteins through weak hydrophobic bonds, making the meat an attractive orange–red color [51]. The freshness of aquatic products can be evaluated by measuring the changes in the color parameter values of the products during refrigeration, including lightness (L*), red (a*), and yellow (b*).

As shown in Figure 7, the L* and a* values in each group decrease with the storage time extension. In contrast, the b* values show an increasing trend. Our results on L* changes were in accordance with those obtained by Sun et al. when applying gelatin coating to grass carp preservation [52]. On the 2nd day of storage, the L* values of the coated groups were significantly (*p* < 0.05) higher than those of the control group, which might be associated with the gloss of the coating itself affecting the light scattering on the meat surface. In addition, a previous study mentioned that calcium alginate coating could form transparent and bright polymers on the surface of fish fillets, which further proved the results mentioned above [53]. During storage progresses, the L* value of the control group was significantly (*p* < 0.05) lower than that of the others, which was the result of the darkened color of meat after severe spoilage [54].

Although astaxanthin, which affects the color of salmon, showed strong antioxidant activity, it was extremely unstable. With the progress of protein degradation during storage, astaxanthin can be released from the formed protein complex and then oxidized to trigger the color change of fish meat [55]. There are obvious differences (*p* < 0.05) in a* values between the samples treated with the coating with and without nanoliposomes, respectively, which might be the result of the antioxidant properties of LC-EO in liposomes delaying the oxidation of related pigments in fish fillets [56]. Nisar et al. [57] also reported a positive effect of the incorporation of essential oils into the coating on a* value changes of fish fillets.

At the end of the storage period, the b* values in the control and XG groups were significantly different from those of the other groups (*p* < 0.05). The reason for the apparent increase in b* value, namely the yellowness, was suspected to be due to the violent non-enzymatic browning reaction that occurred between the amino group of protein and the aldehydes produced by fish fat oxidation [58]. Therefore, the coatings incorporating nanoliposomes, especially the 1:3 and 1:4 coatings, could inhibit or delay the change of color parameter values to maintain the color stability of the fish meat.

#### 3.4.5. Texture Properties

The determination of texture properties was treated as an imperative tool for assessing the effectiveness of preservation methods [57]. Multiple texture properties values of samples, including hardness, adhesiveness, springiness, cohesiveness, chewiness, and resilience, were significantly decreased (*p* < 0.05) (Figure 8), and the texture quality loss of post mortem in salmon was reported to be associated with the modification of myofibrillar protein by proteases and intensified microbial activity [58]. There were also significant differences (*p* < 0.05) in hardness, adhesiveness, chewiness, and resilience between the coating groups containing LC-EO liposomes and the others, while the differences in springiness and cohesiveness were not significant. In the present study, although no significant differences were identified between the texture parameter values of the coating groups added with different concentrations of liposomes, its value variation process was slower than that of the single XG or distilled water treatment. The above results revealed that the addition of LC-EO did not adversely affect the texture properties of fish meat, and the inhibitory effect of LC-EO on bacterial growth and the ability to inhibit the activity of endogenous enzymes might contribute to the delay of texture loss.

#### 3.4.6. Microbial Analyses

Based on the analysis results of various physical and chemical indicators of salmon under the different coating treatments above, the 1:3 coating with the best quality maintenance capability on salmon was selected for antibacterial application. The TVC changes of salmon during storage are displayed in Figure 9a. The initial TVC of fresh salmon was 2.30 CFU/g, and a significant increase in TVC value was observed in both the control and 1:3 coating-treated groups during the storage period. Compared with a significant inhibitory effect on bacterial growth in the 1:3 coating treatment group, the control group exceeded the acceptable limit value of 5.7018 CFU/g for the total number of colonies indicated by ICMSF [59], reaching 6.70 CFU/g on the 6th day.

In Figure 9b, the results show the population changes of *V. parahaemolyticus* in salmon under 1:3 coating treatment during cold storage for 8 days. The initial population of *V. parahaemolyticus* in salmon was 2.89 CFU/g. With the extension of storage time, the population of *V. parahaemolyticus* in the control group increased and reached 5.99 CFU/g on the 6th day, which exceeded the acceptable limit value of microorganisms, 5.7018 CFU/g [59]. The 1:3 coating group with the population of 4.93 CFU/g was lower than the limit value on the 8th day. Furthermore, after storage at 4 °C for 8 days, a 99.47% reduction in *V. parahaemolyticus* population in salmon samples treated with 1:3 coating was observed. The results indicated that the coating treatment could be a promising preservative method to inhibit the growth of live bacteria and *V. parahaemolyticus* in salmon and maintain the food quality.

## 4. Conclusions

In this study, several xanthan-gum-based edible coatings with the incorporation of LC-EO nanoliposomes were developed for salmon preservation. The 1:3 coating effectively delayed the quality deterioration of salmon by minimizing salmon oxidation and controlling the proliferation of microorganisms (TVC and *V. parahaemolyticus*) within 8 days of storage at 4 °C. In conclusion, the biodegradable and biocompatible active edible coatings developed in this study provide an effective and promising approach for maintaining the quality of perishable ready-to-eat fish products.

## Figures and Tables

**Figure 1 foods-11-01535-f001:**
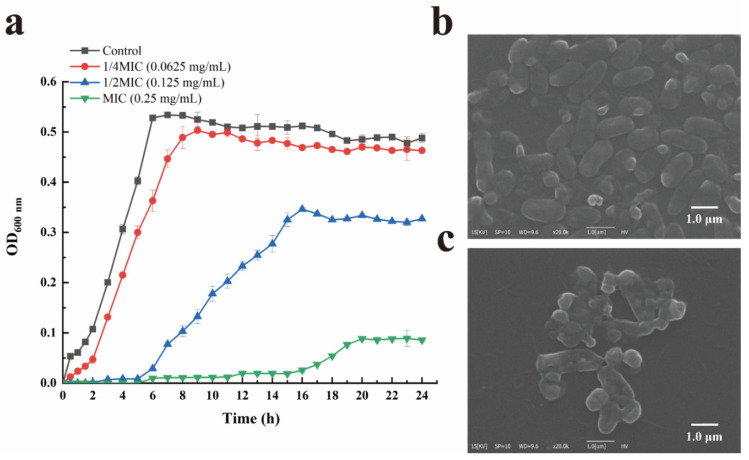
Growth curves of *V. parahaemolyticus* treated with LC-EO at the concentration of 0, 1/4MIC (0.0625 mg/mL), 1/2MIC (0.125 mg/mL), and MIC (0.25 mg/mL), respectively (**a**). SEM images of *V. parahaemolyticus* before (**b**) and after (**c**) 0.5 mg/mL LC-EO treatment. CK: Control group. Scale bar: 1.0 μm.

**Figure 2 foods-11-01535-f002:**
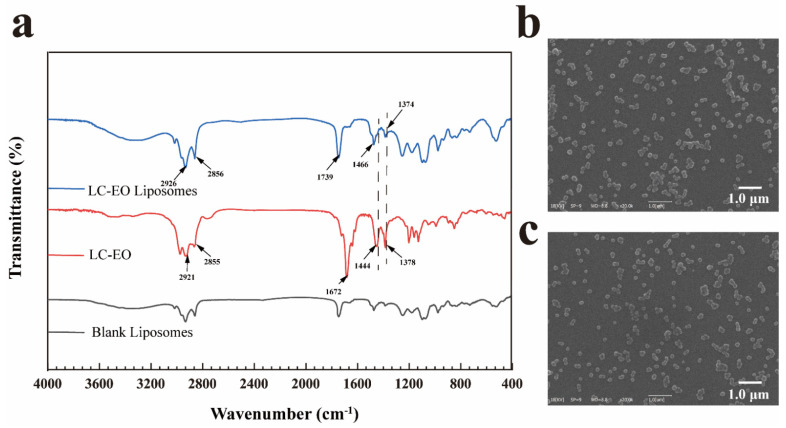
FTIR spectra of blank liposomes, LC-EO, and LC-EO liposomes (**a**). SEM images of blank liposomes (**b**) and LC-EO liposomes (**c**). Scale bar: 1.0 μm.

**Figure 3 foods-11-01535-f003:**
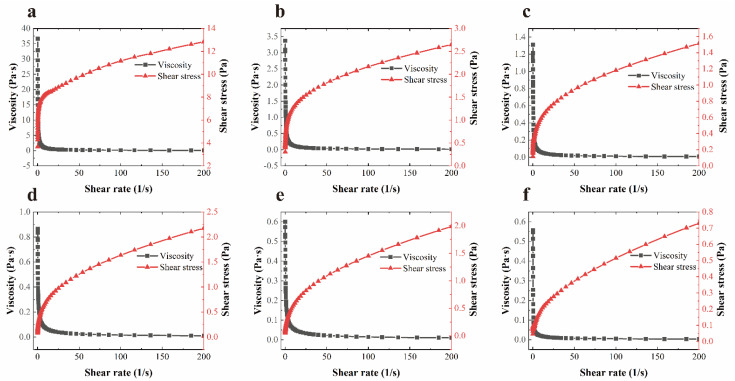
Rheological characterizations of coating solutions at 1:0 (**a**), 1:1 (**b**), 1:2 (**c**), 1:3 (**d**), 1:4 (**e**), 1:5 (**f**) (*v*/*v*). 1:0 = XG solution; 1:1 = XG and liposome (1:1, *v*/*v*) mixed solution; 1:2 = XG and liposome (1:2, *v*/*v*) mixed solution; 1:3 = coated with XG and liposome (1:3, *v*/*v*) mixed solution; 1:4 = XG and liposome (1:4, *v*/*v*) mixed solution; 1:5 = coated with XG and liposome (1:5, *v*/*v*) mixed solution. XG: xanthan gum.

**Figure 4 foods-11-01535-f004:**
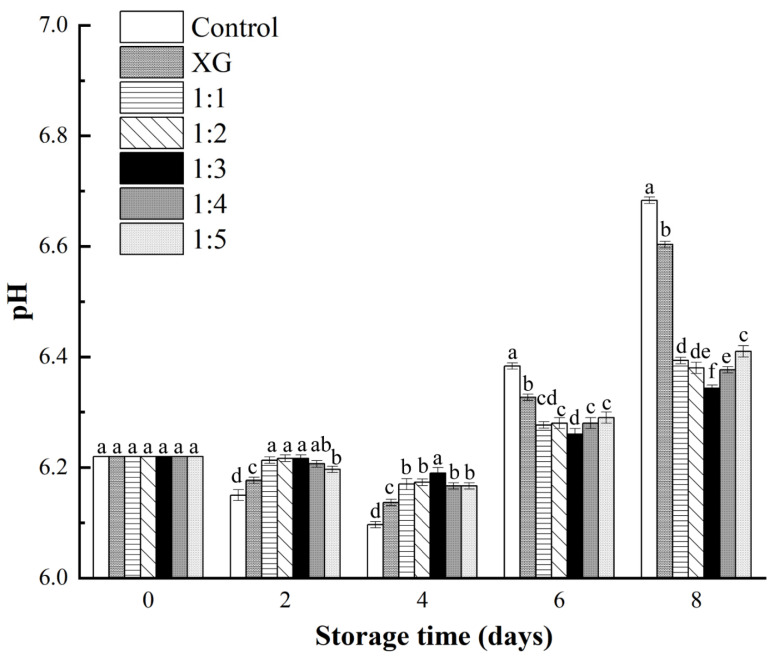
Changes in pH of salmon during storage at 4 °C. Different letter marks on the same day indicate significant difference (*p* < 0.05). Control = coated with distilled water; XG = coated with XG solution; 1:1 = coated with XG and liposome (1:1, *v*/*v*) mixed solution; 1:2 = coated with XG and liposome (1:2, *v*/*v*) mixed solution; 1:3 = coated with XG and liposome (1:3, *v*/*v*) mixed solution; 1:4 = coated with XG and liposome (1:4, *v*/*v*) mixed solution; 1:5 = coated with XG and liposome (1:5, *v*/*v*) mixed solution. XG: xanthan gum.

**Figure 5 foods-11-01535-f005:**
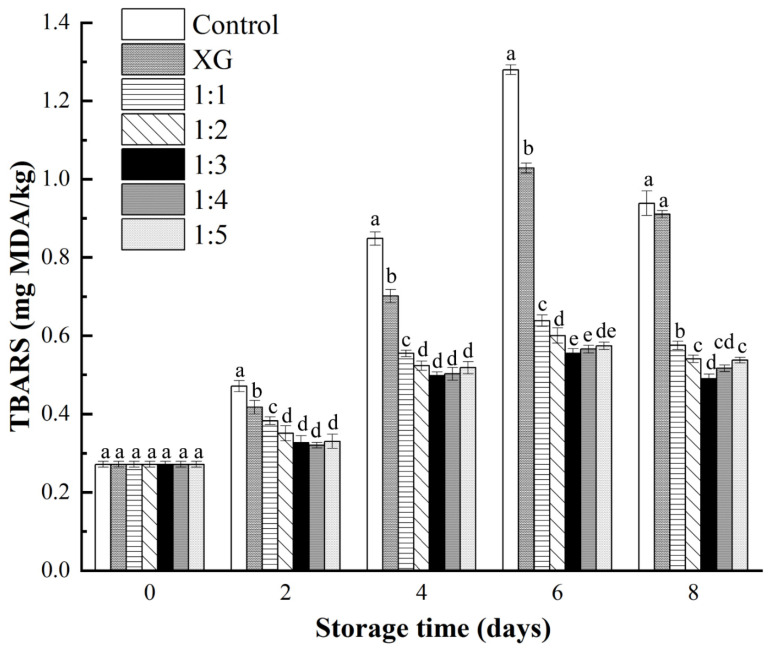
Changes in TBARS values (mg MDA/kg) of salmon during storage at 4 °C. Different letter marks on the same day indicate significant difference (*p* < 0.05). Control = coated with distilled water; XG = coated with XG solution; 1:1 = coated with XG and liposome (1:1, *v*/*v*) mixed solution; 1:2 = coated with XG and liposome (1:2, *v*/*v*) mixed solution; 1:3 = coated with XG and liposome (1:3, *v*/*v*) mixed solution; 1:4 = coated with XG and liposome (1:4, *v*/*v*) mixed solution; 1:5 = coated with XG and liposome (1:5, *v*/*v*) mixed solution. XG: xanthan gum.

**Figure 6 foods-11-01535-f006:**
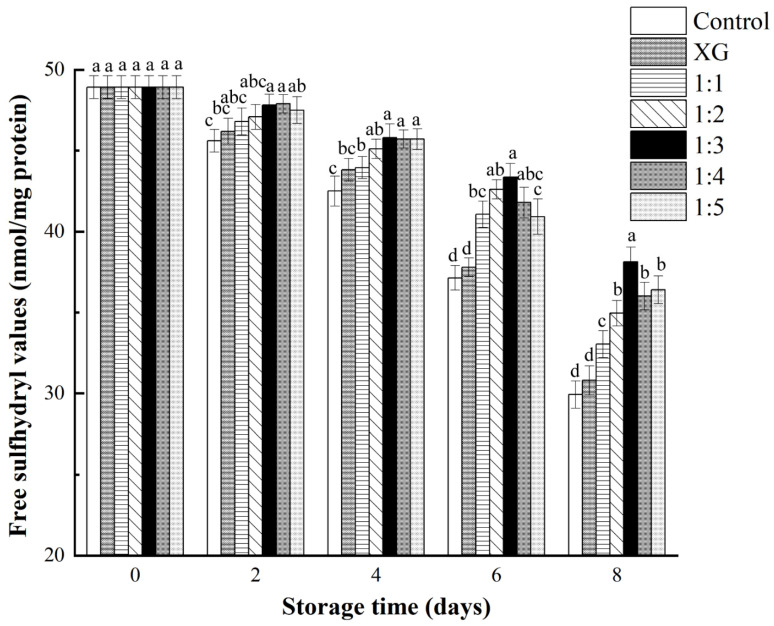
Changes in free sulfhydryl content (nmol/mg protein) of salmon during storage at 4 °C. Different letter marks on the same day indicate significant difference (*p* < 0.05). Control = coated with distilled water; XG = coated with XG solution; 1:1 = coated with XG and liposome (1:1, *v*/*v*) mixed solution; 1:2 = coated with XG and liposome (1:2, *v*/*v*) mixed solution; 1:3 = coated with XG and liposome (1:3, *v*/*v*) mixed solution; 1:4 = coated with XG and liposome (1:4, *v*/*v*) mixed solution; 1:5 = coated with XG and liposome (1:5, *v*/*v*) mixed solution. XG: xanthan gum.

**Figure 7 foods-11-01535-f007:**
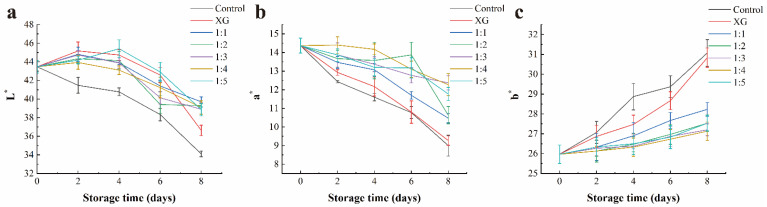
Changes in the color parameter value of salmon: L* (**a**), a* (**b**) and b* (**c**), respectively, during storage. Control = coated with distilled water; XG = coated with XG solution; 1:1 = coated with XG and liposome (1:1, *v*/*v*) mixed solution; 1:2 = coated with XG and liposome (1:2, *v*/*v*) mixed solution; 1:3 = coated with XG and liposome (1:3, *v*/*v*) mixed solution; 1:4 = coated with XG and liposome (1:4, *v*/*v*) mixed solution; 1:5 = coated with XG and liposome (1:5, *v*/*v*) mixed solution. XG: xanthan gum.

**Figure 8 foods-11-01535-f008:**
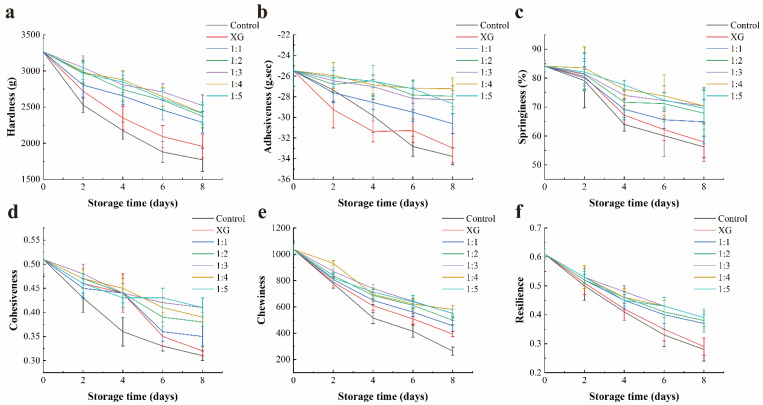
Changes in the texture parameter value of salmon during storage: (**a**) Hardness(g); (**b**) Adhesiveness(g.sec); (**c**) Springiness (%); (**d**) Cohesiveness; (**e**) Chewiness; (**f**) Resilience. Control = coated with distilled water; XG = coated with XG solution; 1:1 = coated with XG and liposome (1:1, *v*/*v*) mixed solution; 1:2 = coated with XG and liposome (1:2, *v*/*v*) mixed solution; 1:3 = coated with XG and liposome (1:3, *v*/*v*) mixed solution; 1:4 = coated with XG and liposome (1:4, *v*/*v*) mixed solution; 1:5 = coated with XG and liposome (1:5, *v*/*v*) mixed solution. XG: xanthan gum.

**Figure 9 foods-11-01535-f009:**
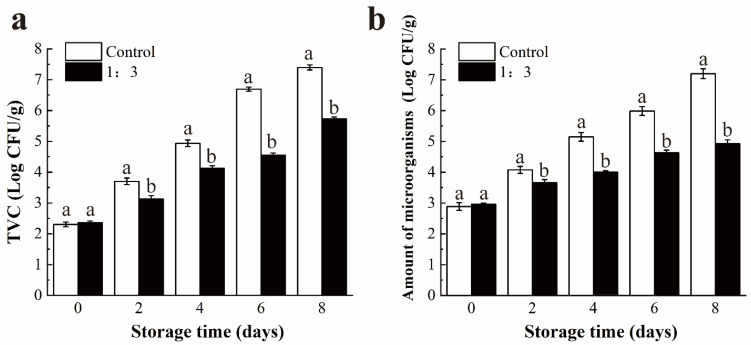
Changes in total viable counts (TVC, Log CFU/g) (**a**) and antibacterial application against *V. parahaemolyticus* of salmon at 4 °C; for 8 days (**b**). Different letter marks on the same day indicate significant difference (*p* < 0.05). Control = coated with distilled water; 1:3 = coated with XG and liposome (1:3, *v*/*v*) mixed solution. XG: xanthan gum.

**Table 1 foods-11-01535-t001:** Characterizations of nanoliposomes with LC-EO at different concentrations.

Parameter	Control	4 mg/mL	5 mg/mL	6 mg/mL	7 mg/mL
Particle Size (nm)	101.27 ± 9.51 ^c^	149.92 ± 9.42 ^b^	168.10 ± 11.97 ^ab^	174.22 ± 10.96 ^a^	185.39 ± 13.46 ^a^
PDI	0.208 ± 0.015 ^c^	0.232 ± 0.021 ^abc^	0.250 ± 0.025 ^ab^	0.254 ± 0.011 ^a^	0.217 ± 0.022 ^bc^
Zeta Potential (mV)	−19.36 ± 0.71 ^d^	−29.28 ± 0.75 ^b^	−32.14 ± 0.52 ^a^	−28.02 ± 0.23 ^c^	−27.07 ± 1.05 ^c^
Encapsulation Efficiency (%)	/	34.2 ± 1.92 ^b^	37.8 ± 1.58 ^a^	31.42 ± 1.96 ^b^	26.60 ± 2.14 ^c^

Results were expressed as the mean ± SD. ^a–d^ Different superscripts in the same row indicated significant differences (*p* < 0.05).

**Table 2 foods-11-01535-t002:** Power–law parameters of coating solutions.

Sample	K (Pa·s)	n	R^2^
XG	6.351 ± 0.073 ^a^	0.122 ± 0.003 ^f^	0.9655
1:1	0.730 ± 0.006 ^b^	0.238 ± 0.002 ^e^	0.9971
1:2	0.302 ± 0.004 ^c^	0.298 ± 0.004 ^d^	0.9951
1:3	0.283 ± 0.002 ^d^	0.383 ± 0.002 ^c^	0.9995
1:4	0.204 ± 0.001 ^e^	0.428 ± 0.001 ^a^	0.9997
1:5	0.077 ± 0.003 ^f^	0.415 ± 0.010 ^b^	0.9842

Results were expressed as the mean ± SD. ^a–f^ Different superscripts in the same column indicate significant differences (*p* < 0.05). XG = XG solution; 1:1 = XG and liposome (1:1, *v*/*v*) mixed solution; 1:2 = XG and liposome (1:2, *v*/*v*) mixed solution; 1:3 = XG and liposome (1:3, *v*/*v*) mixed solution; 1:4 = XG and liposome (1:4, *v*/*v*) mixed solution; 1:5 = XG and liposome (1:5, *v*/*v*) mixed solution. XG: xanthan gum.

## Data Availability

Data is contained within the article.

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
