# Peer review of "Application of Xanthan-Gum-Based Edible Coating Incorporated with Litsea cubeba Essential Oil Nanoliposomes in Salmon Preservation"

_foods, 2022, doi:10.3390/foods11111535_

Round 1
Reviewer 1 Report
The manuscript (Application of xanthan gum-based edible coating incorporated with Litsea cubeba essential oil nanoliposomes in salmon preservation) has good idea. However, it need additional revisions.
1-In the introduction, a paragraph discussing the use of coating to preserve food and inhibit microorganisms should be added. I recommend that you read this article ((Production of biodegradable film from soy protein and essential oil of lemon peel and use it as cheese preservative. https://doi.org/10.37077/25200860.2017.40.
2-Materials and Methods, several work methods lack scientific references, making it difficult to validate the method's accuracy like Determination of particle size, PDI, zeta potential, and EE of LC-EO method, Fourier transform infrared spectroscopy (FTIR) method and.......etc. . I suggest some references.
3-Page 2 line 73, Antibacterial activity method is unclear, How is the count of bacteria inhibited?
4-Page 7 figure 2, Theses (B and C) images are unclear.
5-Page 15, figure 9b, The word (population) is not appropriate with microorganisms, it must be replaced with another suitable word such as microorganisms.
6-Rewrite the manuscript's conclusions because it contains a lot of results.
Reviewer 2 Report
The study is not entirely novel given that Litsea cubeba essential oil (LC-EO) extracts have already been used as antimicrobials in other foods. However, for the first time this work provides a study on the incorporation of nanoliposomes with LC-EO extracts on xanthan coatings for preserving salmon.
In general, it can be observed that the introduction deals with easy-to-understand information which draws the reader in. With the exception of a few parts, the methodology section is also well written. Furthermore, most of the results are discussed correctly, while the conclusions are accurate and concrete.
However, it is necessary to review and correct the following details:
The authors need to review the English language used in the manuscript.
Line 79: Provide the meaning of OD (optical density)
Lines 128-129: Describe what each treatment consists of
Line 133: It is recommended starting this sentence with a word or connector. Apply this to the whole document.
Line 168: This line cannot be understood and needs restructuring.
Lines 183-186: The type of test used to compare the averages is not mentioned.
Lines 191-192: This information cannot be understood. Please mention which treatments tended to be stable.
Lines 207-211: In the context in which the information relating to particle size of the treatments with LC-EO is mentioned, is the 101.27 nm value correct? Bearing in mind that this is the negative control, it should be starting from 4 mg/ml (149.92) in my opinion.
Line 211: In which treatment is the value 0.30 observed
Line 268: Does the pH increase or decrease?
Line 291: Is the 1-2 mg MDA limit specific to just fish?
Line 353: Mention in concrete terms what Sun and the other collaborators investigated.
Figure 1: Mention at which concentration the legend “MIC” refers to.
Reviewer 3 Report
I reviewed the manuscript entitled, Application of xanthan gum-based edible coating incorporated with Litsea cubeba essential oil nanoliposomes in salmon preservation. The manuscript is well written with appropriate scientific literature. The Methodology section should be improved with more details. Based on these observations, I would recommend major revision.
Line 37: Is the problem related to Asia? What about Europe and Africa?
Line 48: few lines about Nanoliposome technology should be introduced
2.3.2.1. Determination of particle size, PDI, zeta potential, and EE of LC-EO: describe the methodology with appropriate literature.
2.3.2.2. Fourier transform infrared spectroscopy (FTIR): describe the methodology
2.5. Rheological properties of the coating solutions: provide the detailed methodology
Line 133: sentence should not start with number. Please revise as sample sample (5 g)…….
2.6.6. Texture evaluation: describe the methodology
Line 199: scientific name must be in Italics
Figure 7: improve the quality, especially, axis
Figure 9: please perform statistical analysis
Lines 388-390: is this true? Authors did not conduct any statistical analysis
Line 321: exhibited…. Replace with illustrated or showed
Why authors selected 1:3 for microbial analysis. As the authors mentioned, it was based on physical and chemical indicators of salmon. Which chemical and physical parameters?
3.2. Characterization of LC-EO liposomes: more discussion should be needed.
Why 3.4.1. pH increased in 1:4 on day 8
None of the references are according to journal format
Round 2
Reviewer 1 Report
Dear Editor (S)
Authors did all necessary changes to improve the manuscript Authors did all necessary changes to improve the manuscript but I detected inappropriate self-citations by authors. More than 6 authors' self-citations.
Reviewer 3 Report
Authors are now answered the questions raised by me. In my opinion, this version of the manuscript can be accepted for publication in Foods.